# Upper Mantle Velocity Structure Beneath the Yarlung–Tsangpo Suture Revealed by Teleseismic P-Wave Tomography

Dong Yan [1], You Tian [1,2,*], Zhiqiang Li [1] and Hongli Li [1]

1 College of Geoexploration Science and Technology, Jilin University, Changchun 130026, China; yandong20@mails.jlu.edu.cn (D.Y.); zhiqiang21@mails.jlu.edu.cn (Z.L.); lihongli@jlu.edu.cn (H.L.)

2 Changbai Volcano Geophysical Observatory, Ministry of Education, Jilin University, Changchun 130026, China

* Correspondence: tianyou@jlu.edu.cn

**Abstract:** We applied teleseismic tomography to investigate the 3D P-wave velocity (Vp) structure of the crust and upper mantle at depths of 50–400 km beneath the Yarlung–Tsangpo suture (YTS), by using 6164 P-wave relative travel-time residuals collected from 495 teleseismic events recorded at 20 three-component broadband seismograms. A modified multi-channel cross-correlation method was adopted to automatically calculate the relative arrival-time residuals of all teleseismic events, which significantly improved the efficiency and precision of the arrival-time data collection. Our results show that alternating low- and high-Vp anomalies are visible beneath the Himalayan and Lhasa blocks across the YTS, indicating that strong lateral heterogeneities exist beneath the study region. A significant high-Vp zone is visible beneath the southern edge of the Lhasa block at 50–100 km depths close to the YTS, which might indicate the rigid Tibetan lithosphere basement. There exists a prominent low-Vp zone beneath the Himalayan block to the south of the YTS extending to ~150 km depth, which might be associated with the fragmentation of the underthrusting Indian continental lithosphere (ICL) and induce localized upwelling of asthenospheric materials from the upper mantle. In addition, significant low-Vp anomalies were observed beneath the Yadong–Gulu rift and the Cona–Sangri rift extending to ~300 km depth, indicating that the tearing of the subducted ICL might provide pathways for the localized asthenospheric materials upwelling, which contributes to the widespread distribution of north–south trending rifts and geothermal activities in southern Tibet.

**Keywords:** teleseismic tomography; P-wave velocity structure; southern Tibet; Indian continental lithosphere; rifting system

## 1. Introduction

The Indian continental lithosphere (ICL) has been subducted northward beneath the Eurasian continent since the closure of the Tethys Ocean in the early Cenozoic (~50 Ma), contributing to the large-scale crustal shortening and thickening, which led to the formation of the Tibetan Plateau [1,2]. Several geodynamic mechanisms have been proposed to explain the tectonic evolution of the Tibetan Plateau, such as the extrusion of tectonic blocks along large strike-slip faults [3], the crustal shortening and thickening [4], crustal channel flow [5] and the southward subduction of the Eurasian plate [6]. However, there is still no consensus on the mechanism of the extensive post-collision convergence between the Indian and Eurasian plates, which could reach nearly 2000 km [1,7]. In addition, the geological feature of the Himalayan–Tibetan orogen is characterized by varieties of active fault systems and widespread magmatism [8–12]. Therefore, the Himalayan–Tibetan orogen has become a natural laboratory in which to study the continent–continent collision processes due to its complex lithospheric structure and intense crustal deformation.

The northward subduction of the Tethys Oceanic slab along the southern margin of the Eurasian plate in the late Cretaceous led to the formation of an Andes-type magmatic

arc. Then, the slab rollback and back-arc extension could have resulted in widespread post-collisional potassic volcanism at ~10–26 Ma [12–14]. The Indian–Eurasian plates' convergence has contributed to the formation of several major tectonic blocks and suture zones in the central and southern Tibetan Plateau (Figure 1). The Himalayan block is situated between the Indian Shield to the south and the Yarlung–Tsangpo suture (YTS) to the north, consisting of the Tethyan Himalaya, Higher Himalaya, and Lesser Himalaya. The YTS delineates the boundary between the Indian continental sediments and the Eurasian continental rocks after the closure of the Tethys Ocean [14]. The Lhasa, Qiangtang, and Songpan–Garze blocks are distributed from south to north, separated by the Bangong–Nujiang suture (BNS) and the Jinsha River suture (JRS), respectively [1].

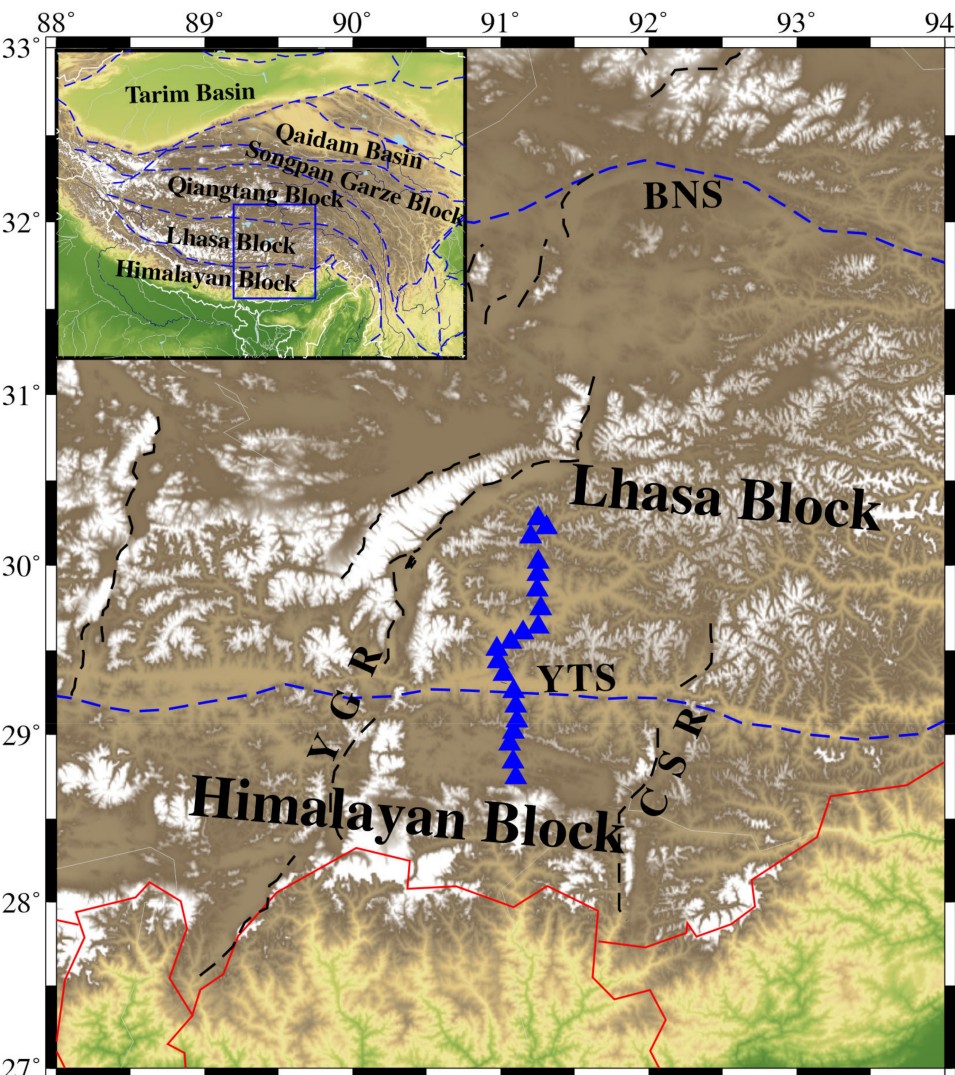

**Figure 1.** Map view of the surface topography and major tectonic settings of southern Tibet. The blue triangles denote seismic stations used in this study. The red solid lines depict the international boundary. The blue dashed lines denote sutures separating major tectonic blocks. The black dashed lines depict major north–south trending rifts. YTS: Yarlung–Tsangpo suture; BNS: Bangong–Nujiang suture; YGR: Yadong–Gulu rift; CSR: Cona–Sangri rift.

The southern part of the Tibetan Plateau is characterized by active north–south (N–S) trending rifting systems associated with the east–west extension, such as the Tangra Yum Co rift (TYR), Pumqu–Xianza rift (PXR), Yadong–Gulu rift (YGR), and Cona–Sangri Rift (CSR) from west to east, which are roughly perpendicular to the Indian and Eurasian plates' convergence direction [15]. Although some models, such as the gravitational collapse,

delamination, and fragmentation of the Indian lithosphere [4,16–18], have been proposed, there is still no consensus on the formation mechanism of the N–S trending rifts to date.

The Tibetan Plateau is characterized by remarkable crustal thickness which is nearly double the thickness of the average continental crust [6]. Previous receiver function studies show that the crustal thickness exhibits significant discrepancy on both sides of the YTS which decreases from ~80 km beneath the Lhasa block to ~50 km beneath the Himalayan block. Furthermore, the crustal thickness beneath the Qiangtang and Songpan–Garze blocks is relatively thinner (~70 km) in comparison to the Lhasa block [19–21]. The Indian lithosphere underthrusting model has been proposed to explain the large-scale crustal shortening and thickening of the Tibetan Plateau, namely that the upper crust of the Indian plate was stripped off to form the Himalayan block and the lower crust subducted further northward beneath the Lhasa block [22]. However, the angle and front of the underthrusting Indian lithosphere still remain debated. Some previous studies suggest that the ICL has underthrusted ~150 km north of the YTS [19,23,24], while other studies argue that the front of the underthrusting Indian plate has extended beyond the YTS and might be located near the BNS beneath the central Tibet [25,26].

The majority of the seismological studies before the 1990s were limited to the large-scale deep structures beneath the Tibetan Plateau due to sparse seismic station distributions. Since the deployment of seismic arrays by international collaborative projects such as PASS-CAL, INDEPTH, and HiCLIMB during the last few decades, many investigations have been conducted to obtain further insight into the evolution processes and small-scale deep structures of the Himalayan–Tibetan orogen [27–33]. In addition, much seismological research has been conducted in the Tibetan Plateau over the past decade, such as deep seismic soundings [34–37], seismic tomography [38–46], and teleseismic receiver functions [47–51]. However, most of the previous studies paid much more attention to southeastern Tibet and there are few studies on the detailed 3D velocity structure of the upper mantle beneath the YTS due to the limitations of the available seismic data.

In the present study, we apply teleseismic tomography [52] to determine the high-resolution 3D P-wave velocity (Vp) structure of the upper mantle beneath the Himalayan and Lhasa blocks across the YTS extending to ~400 km depth. Our results reveal the seismological characteristics beneath major tectonic blocks in the study region and shed new light on the tectonic evolution process of the Himalayan–Tibetan orogen.

## 2. Data and Method

We deployed a 200 km long linear seismic array from October 2017 to June 2019 with ~10 km average station interval, which is nearly perpendicular to the YTS from the northern part of the Himalayan block to the southern part of the Lhasa block (Figure 1). The seismic array consists of 20 sets of Güralp CMG-3ESP seismometers and Reftek-130 digital recorders. We selected teleseismic events compiled at the United States Geological Survey (USGS) earthquake catalog, with epicentral distances between 30° and 90°, which have magnitudes greater than M 5.0. In addition, each event was recorded by at least 6 seismic stations. The vertical component waveform data were preprocessed by removing the mean values and linear trends. Then, all seismograms were cut off with a time window of 20 s before and 100 s after the P-wave arrivals and carefully checked with clear P-wave onsets. As a result, 495 high-quality teleseismic events were selected for the tomographic inversion. Figure 2 shows that the teleseismic events generally exhibited good azimuthal coverage around the study region, and the majority of the teleseismic events were distributed along the Pacific subduction zone to the east.

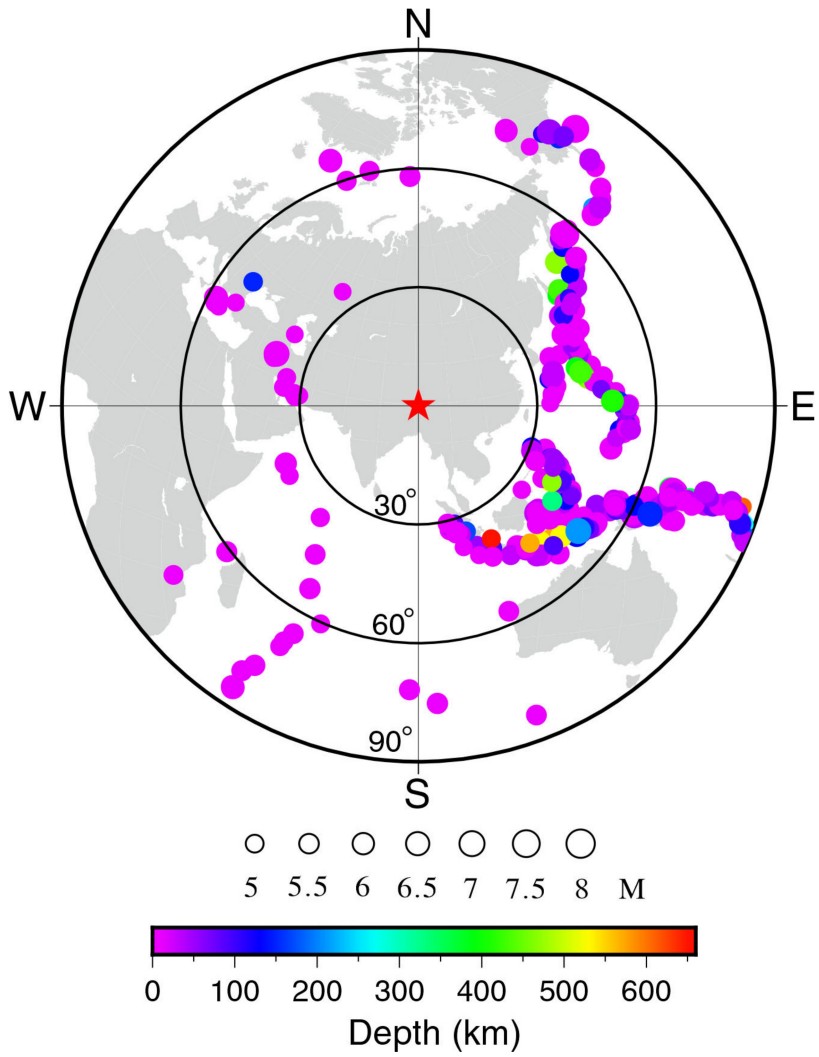

**Figure 2.** Epicentral distribution of the 495 teleseismic events (color circles) used in this study. The red star denotes the center of the study region. The size and color of the circles denote the earthquake magnitude and the focal depth, respectively, whose scales are shown at the bottom.

We used the teleseismic tomography method [52] to investigate the 3D upper-mantle Vp structures down to ~400 km depth beneath the study region. In order to reduce the effects of the heterogeneities in the crust and uppermost mantle, we used the relative travel-time residuals instead of the absolute travel-time residuals of the teleseismic events in the tomographic inversion [52]. We adopted a modified multi-channel cross-correlation (MMCC) method [53] to calculate the relative arrival-time residuals of all teleseismic waveforms automatically (Figure 3). The method combined the phase weight stacking scheme and the geometric normalized cross-correlation method [54,55], which significantly improved the precision of the relative travel-time residuals calculation, especially for those teleseismic events with relatively low signal-to-noise ratios. Figure 3B,C shows 16 teleseismic waveforms aligned based on their theoretical P-wave arrival times and common seismic phases, respectively. As a result, we obtained 6164 relative travel-time residuals collected from 495 high-quality teleseismic events. The average relative-time residual at each seismic station was calculated according to the azimuth range in four different quadrants (Figure 4A–D). Then, we obtained a mean relative residual by averaging all relative travel-time residuals at each station which is shown in Figure 4E. There were delayed arrivals at stations in the Himalayan block to the south of the YTS, and early arrivals were visible at stations on the southern edge of the Lhasa block close to the YTS. In contrast, the

stations of delayed arrivals were distributed in the northern part of the Lhasa block, which indicates the lateral heterogeneities beneath the study region. The main characteristics of the average relative travel-time residual distributions were generally consistent with the P-wave velocity anomalies revealed by previous studies [17,18], indicating that our teleseismic travel-time data had a good quality.

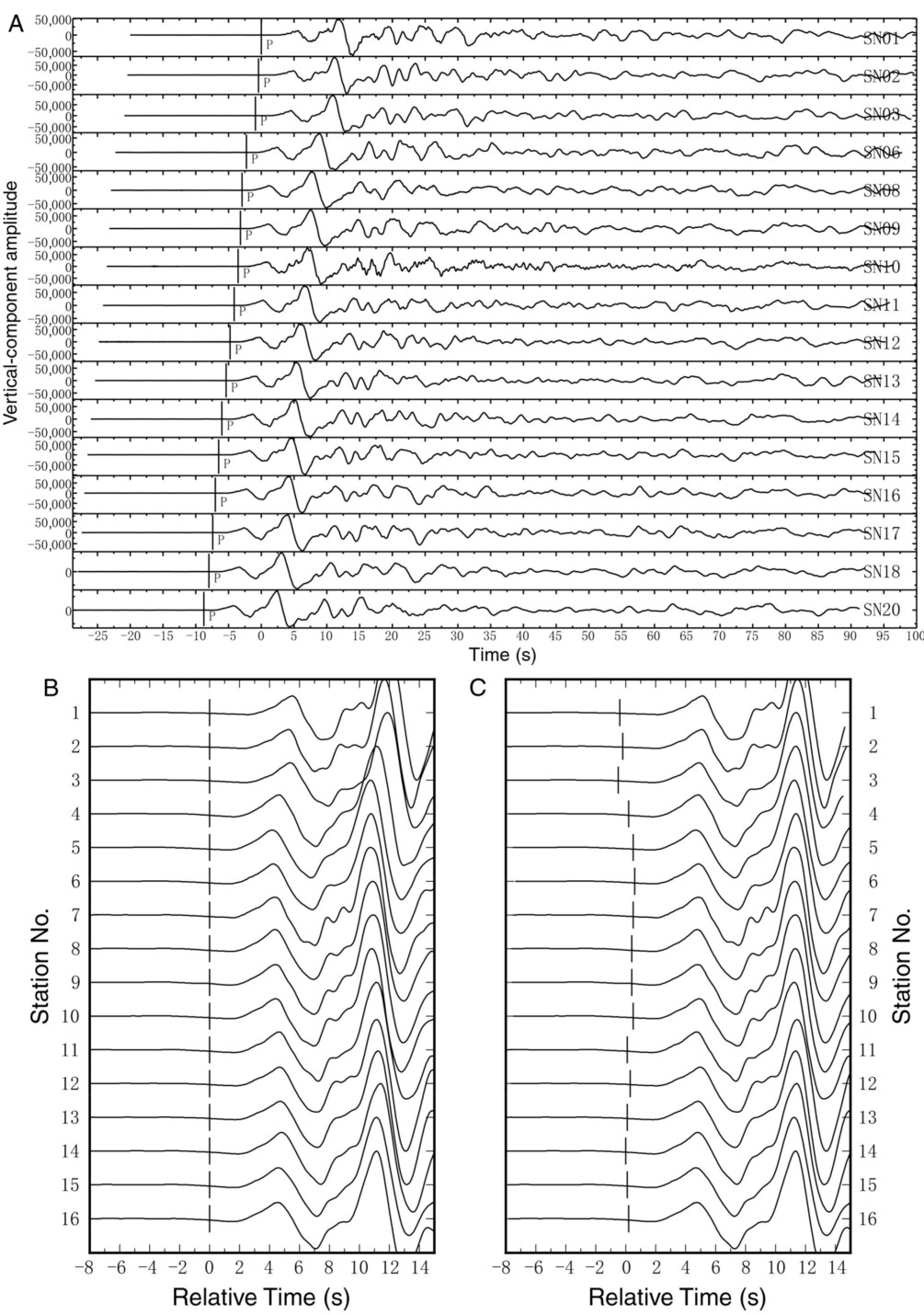

**Figure 3.** (**A**) An example showing vertical-component waveforms from a teleseismic event that occurred on 20 December 2018. (**B,C**) Comparison of the teleseismic waveforms aligned based on their theoretical P-wave arrival times and common phases. The vertical lines in each panel denote theoretical P-wave arrival times.

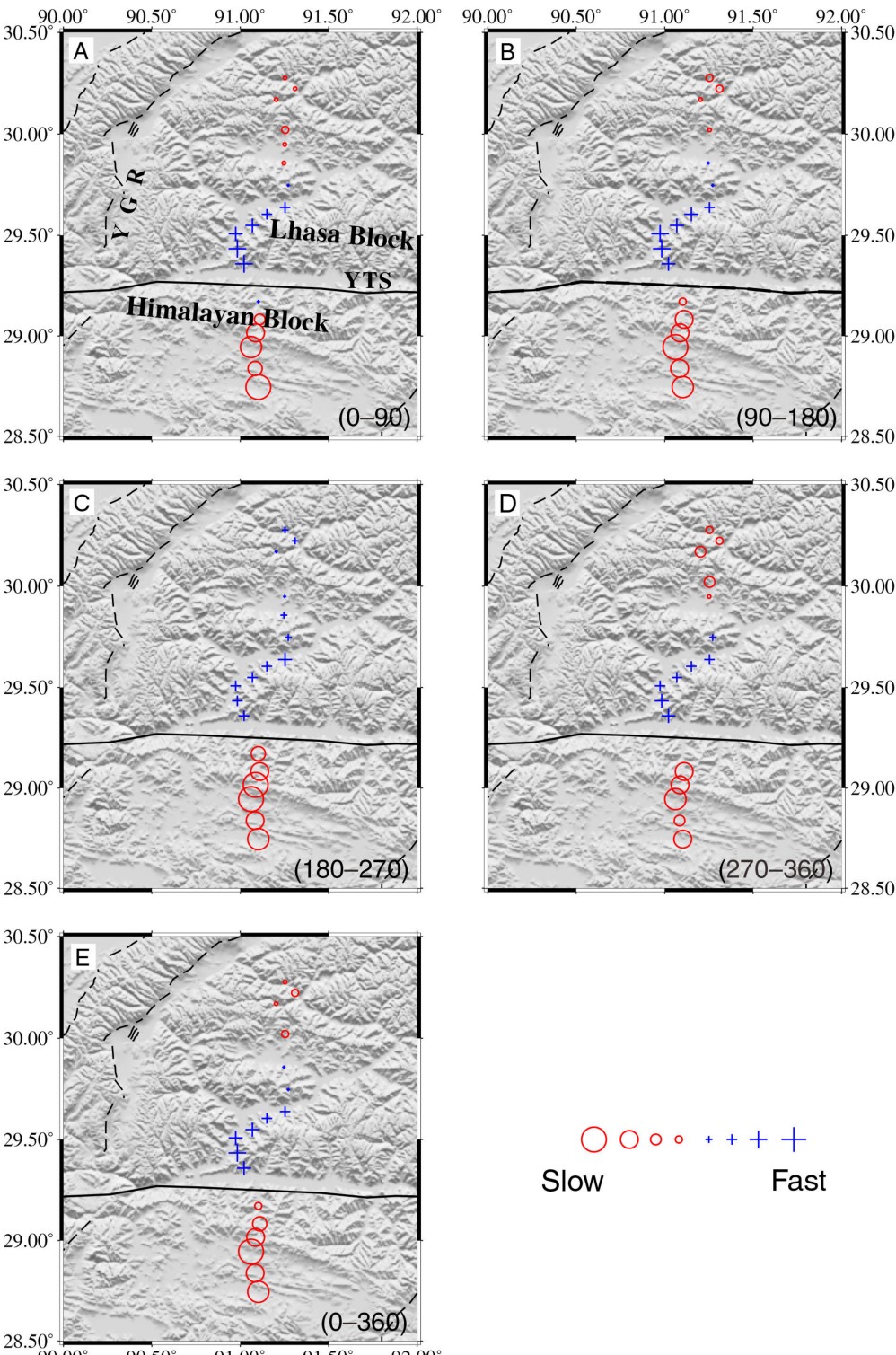

**Figure 4.** (**A–D**) Distributions of average relative travel-time residuals at each station in four different source quadrants. (**E**) Average relative travel-time residuals of all teleseismic events used in this study. The red circle and blue cross symbols denote delayed and early relative travel-time residuals, respectively, whose scale is shown at the bottom. The black solid line depicts the Yarlung–Tsangpo suture (YTS). The azimuth range of teleseismic events is shown at the lower-right corner of each map. Other labels are the same as those in Figure 1.

The modified iasp91 velocity model [56] was adopted as the 1D reference velocity model to calculate the theoretical P-wave travel times (Supplementary Figure S1). We set up 3D grid nodes beneath the study region. The lateral grid interval was set as 0.5° and, in the vertical direction, the grid nodes were set up at −100, 50, 100, 150, 200, 300, 400, 500, 600, and 800 km depths after making resolution tests with different grid intervals. The Conrad and Moho discontinuities were taken into consideration which were set at 20 and 60 km depths, respectively. We calculated the ray paths and theoretical travel times by using an efficient 3D ray tracing method [57] that combined the pseudo-bending method with Snell's law and applied to the continuous medium and the velocity discontinuities, respectively. The damped LSQR algorithm [58] was adopted to solve the large and sparse system of observation equations. The damping parameter was used to constrain the amplitude of sharp velocity perturbations in small-scale structures. We used the trade-off curve between the norm of the tomographic models and the root-mean-square (RMS) travel-time residuals by performing multiple inversions to determine the optimal damping parameter as 40 (Figure 5). The final RMS travel-time residual reduced to ~0.19 s after the tomographic inversion.

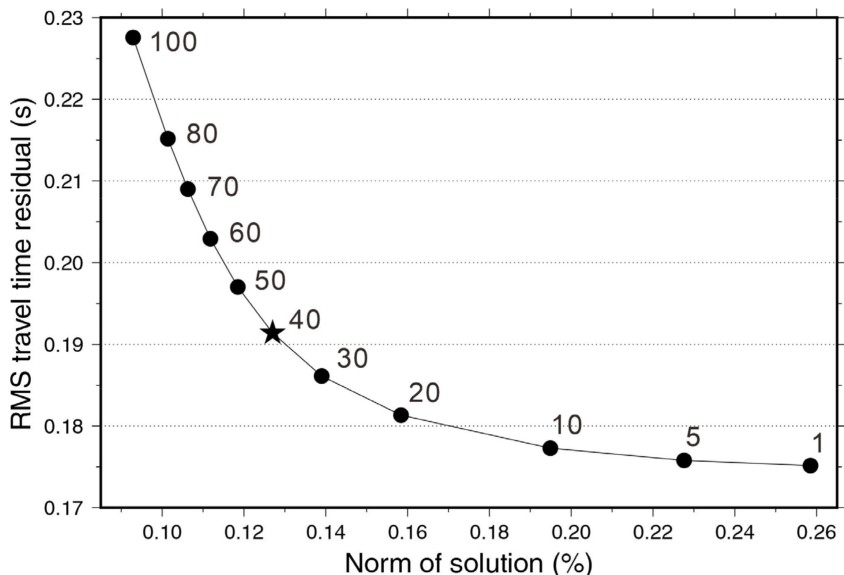

**Figure 5.** Trade-off curve for the norm of tomographic models and RMS travel-time residuals of the inversion with different damping parameters. The number close to each circle represents the value of the damping parameter. The star symbol denotes the optimal damping parameter selected in this study.

## 3. Results and Discussion

### 3.1. Ray Path Coverage

The hit counts' distributions at six representative depths in the lower crust and upper mantle, as shown in Figure 6, indicate the number of seismic rays passing around each grid node and can be used to measure the ray coverages beneath the study region. The ray coverage was relatively good at depths of 50–300 km, especially beneath the central and eastern parts of the study region. The ray coverage was generally low beneath the marginal areas of the study region due to the sparse station density. In addition, the ray coverage density was relatively low beneath the western part of the study region at ~400 km depth because most of the teleseismic events occurred in the western Pacific subduction zone to the east. In general, the ray coverage density beneath the study region was good enough to be applied for the tomographic inversion.

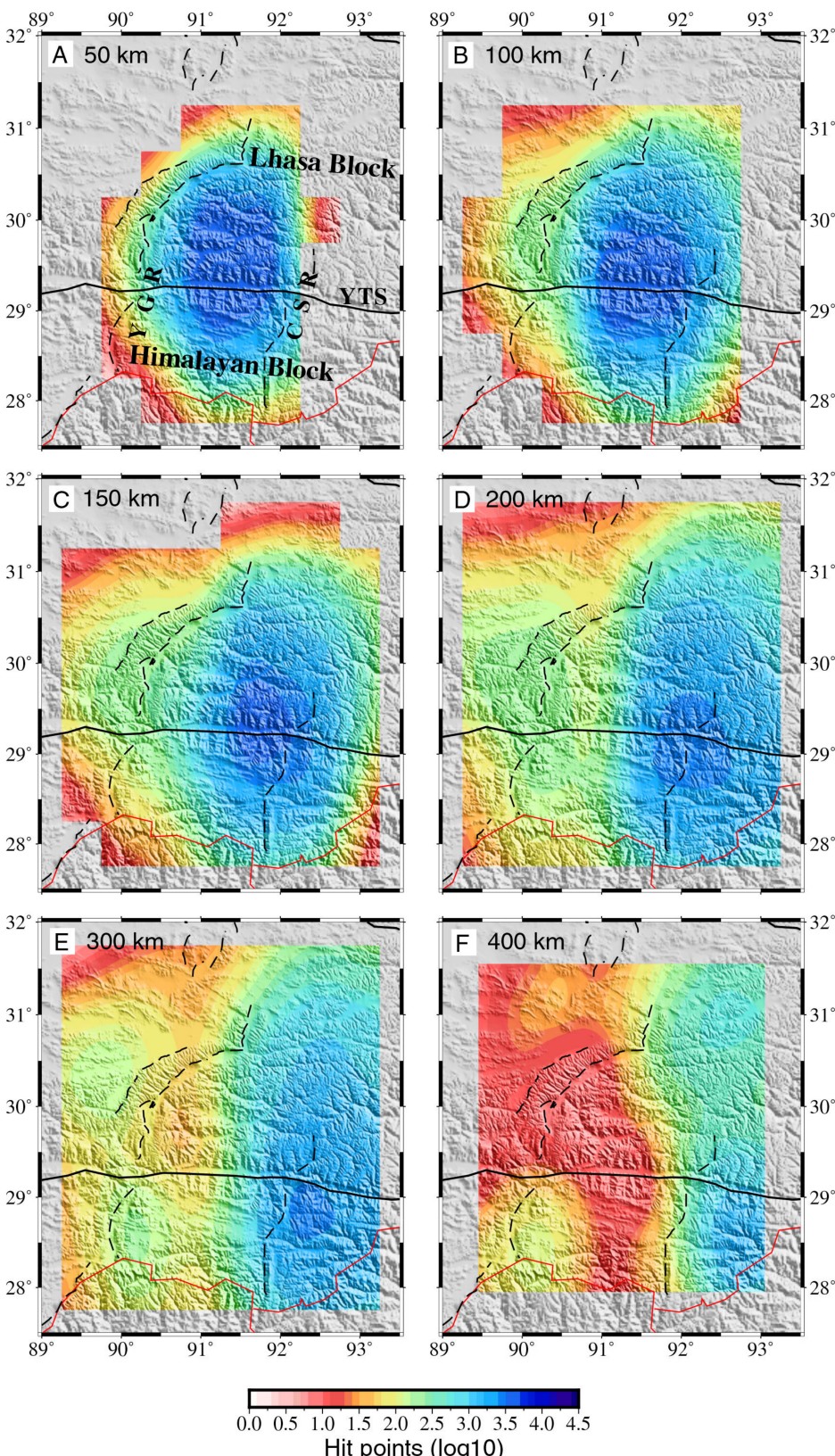

**Figure 6.** (**A–F**) Distribution of hit counts at six representative depths beneath the study region, whose scale is shown at the bottom. The layer depth is shown at the upper-left corner of each map. Other labels are the same as those in Figure 4.

### 3.2. Checkerboard Resolution Tests

In this study, we conducted the checkerboard resolution test (CRT) to assess the reliability of our 3D Vp models. The positive and negative velocity perturbations of 3% relative to the 1D reference velocity model were assigned alternately to the 3D grid nodes in both the horizontal and vertical directions. We computed the synthetic travel times for the checkerboard model with the same ray paths as the observed travel-time data. In addition, we conducted the CRT by adding random noise with a standard deviation of 0.1 s to the synthetic arrival-time data to simulate the picking errors in the observed arrival times. We conducted the CRT with 0.35° (~35 km), 0.5° (~50 km) and 0.75° (~75 km) lateral grid intervals to determine a suitable grid spacing for inversion (Figure 7, Supplementary Figures S2 and S3). Figure 7 shows the CRT results at six representative depths with a lateral grid interval of 0.5°. The CRT results show that the Vp perturbations model can be well resolved beneath the central part of the study region at shallow depths (50–100 km), albeit the resolution is relatively low beneath the marginal areas due to the sparse seismic ray coverage there. At depths of 150–200 km, the resolution of the Vp model was relatively good, especially in the central and eastern parts of the study region. The Vp model with a 0.5° lateral grid interval was resolved better than that with 0.35° and 0.75° lateral grid intervals. However, the resolution reduced slightly at ~300 km depth, which may have been caused by the relatively sparse ray coverage at that depth. The Vp model can only be resolved in the eastern part of the study region at ~400 km depth, which may be related to the concentrated distribution of teleseismic events that occurred along the western Pacific subduction zone, resulting in relatively low ray coverage density in the western part of the study region. After comparing all CRT results with different grid intervals, we chose the optimal lateral grid interval as 0.5° × 0.5° in the tomographic inversion. In general, the CRT results show that the Vp model can be well recovered beneath the majority of the study region with good ray coverage at depths above ~400 km.

### 3.3. Tomographic Imaging

Figure 8 shows map views of our 3D Vp model of the upper mantle at six representative depths beneath the study region, which are clipped based on the hit points distribution (>30) and only the high-resolution areas are displayed. Figure 9 shows six vertical cross-sections of the 3D Vp model along the longitude direction. Our results show that alternating low- and high-Vp anomalies are visible beneath the study region, which indicates that there exists strong lateral heterogeneities beneath the Himalayan and Lhasa blocks across the YTS. The anomalies of our Vp model were spatially correlated with major tectonic blocks and active rifting zones. There exists an obvious velocity contrast on both sides of the YZS at depths of 50–100 km (Figures 8A and 9C,D). A significant low-Vp zone (LVZ) was visible beneath the Himalayan block to the south of the YTS and there exists a prominent high-Vp zone (HVZ) beneath the southern edge of the Lhasa block close to the YTS. The boundary between two velocity anomaly zones was delineated distinctly by the YTS, which exhibited a much clearer characteristic in comparison to previous tomographic results [16–18]. In addition, relatively significant low-Vp anomalies were observed beneath the YTS extending to ~200 km depth (Figure 9A,B) and there exist high-Vp anomalies at 300–400 km depths beneath the YTS (Figure 8E,F and Figure 9A–C). Significant low-Vp anomalies were visible in the upper mantle extending to ~300 km depth beneath the YGR and the CSR (Figure 8B–E and Figure 9B–E). These results are in good agreement with previous body-wave tomographic results [17,18], and exhibit a higher lateral resolution in comparison to the Vp structures investigated by previous studies [16,26].

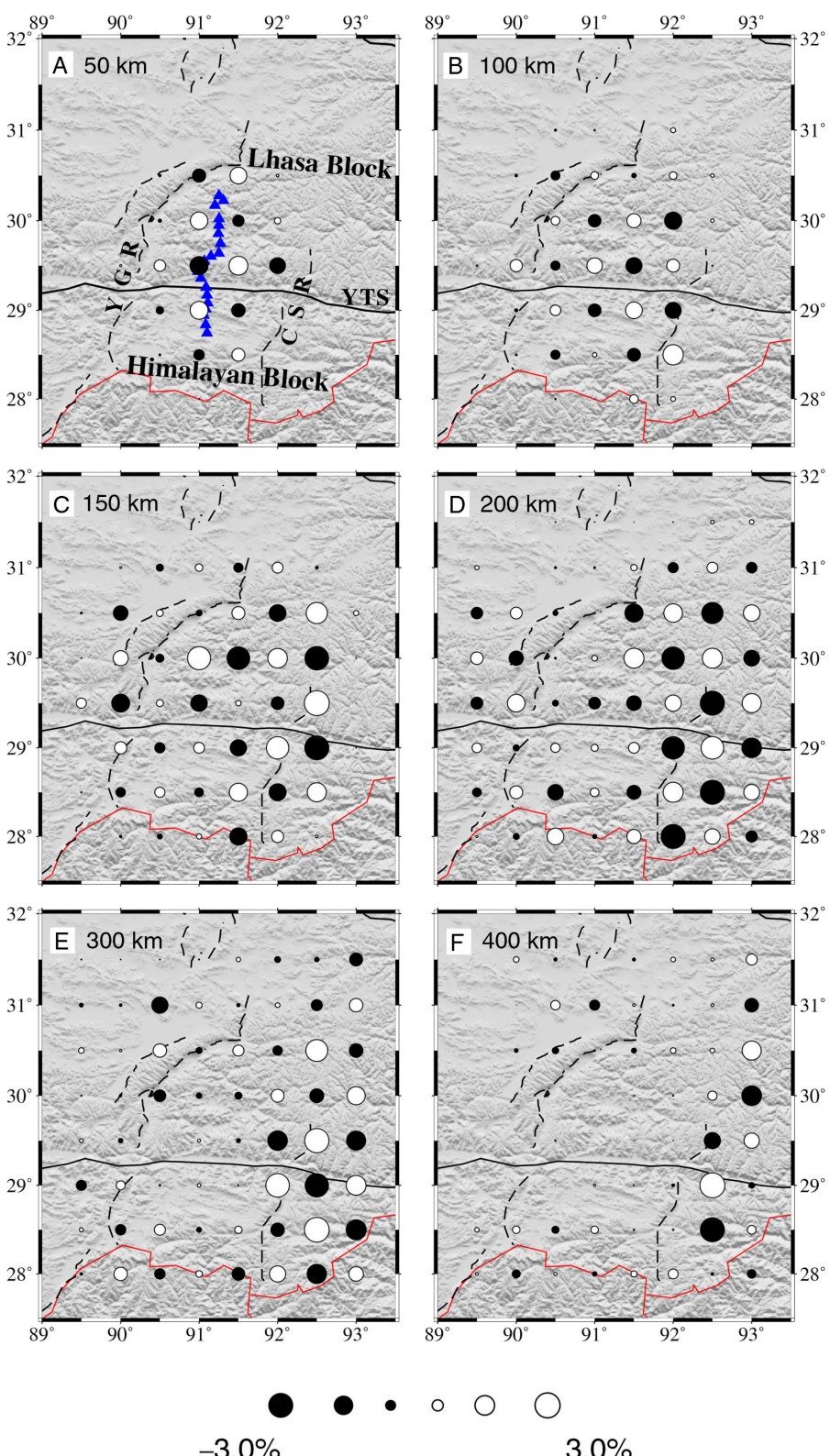

**Figure 7.** (**A**–**F**) Map views showing results of a checkerboard resolution test (CRT) with a lateral grid interval of 0.5°. The layer depth is shown at the upper-left corner of each map. The blue triangles denote seismic stations used in this study. The open and solid circles denote positive and negative Vp perturbations (%), respectively, whose scale is shown at the bottom. Other labels are the same as those in Figure 4.

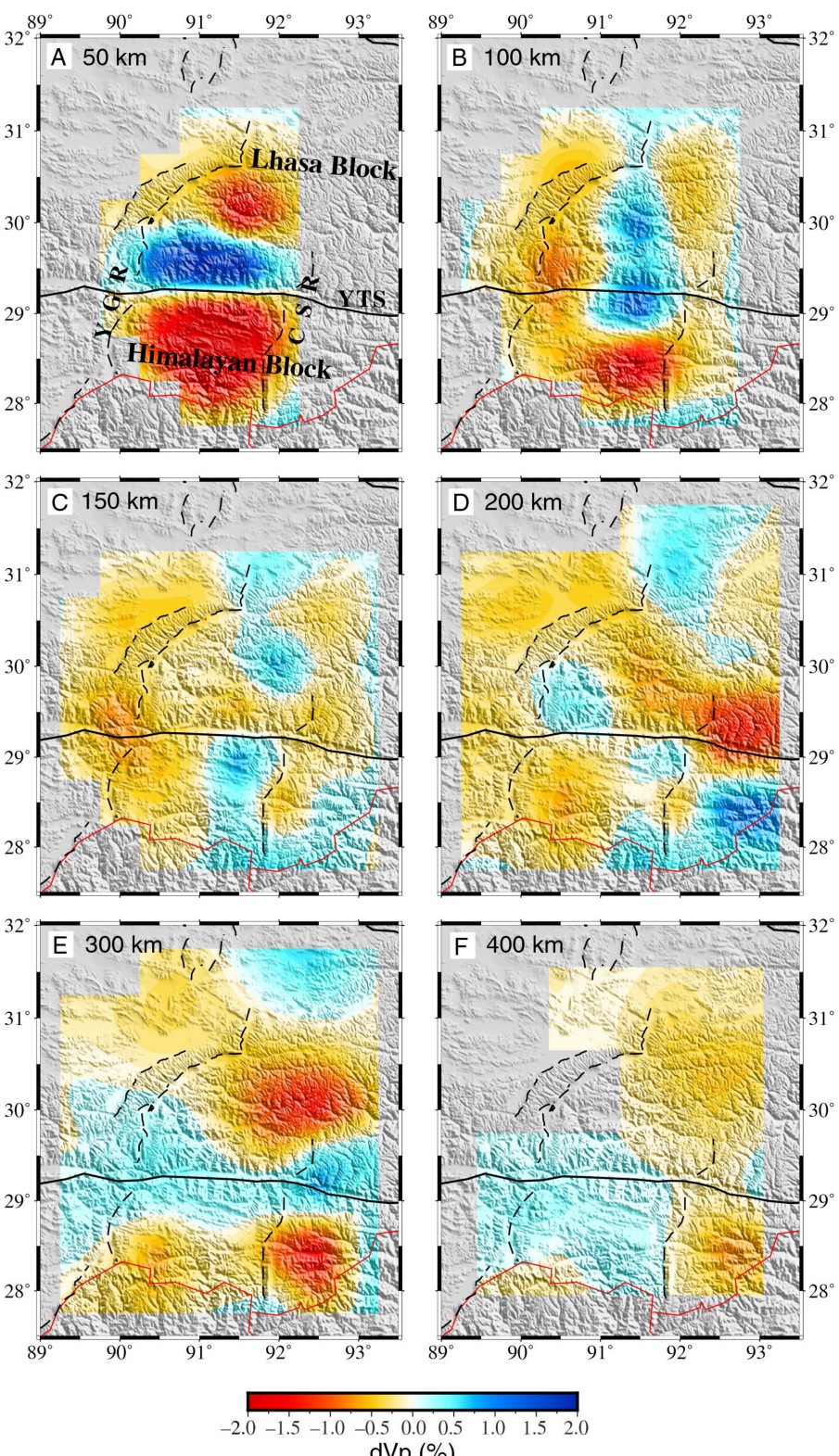

**Figure 8.** (**A–F**) Map views showing the 3D P-wave velocity (Vp) model at six representative depths. The layer depth is shown at the upper-left corner of each map. The red and blue colors represent low and high Vp perturbations, respectively, whose scale is shown at the bottom. Other labels are the same as those in Figure 4.

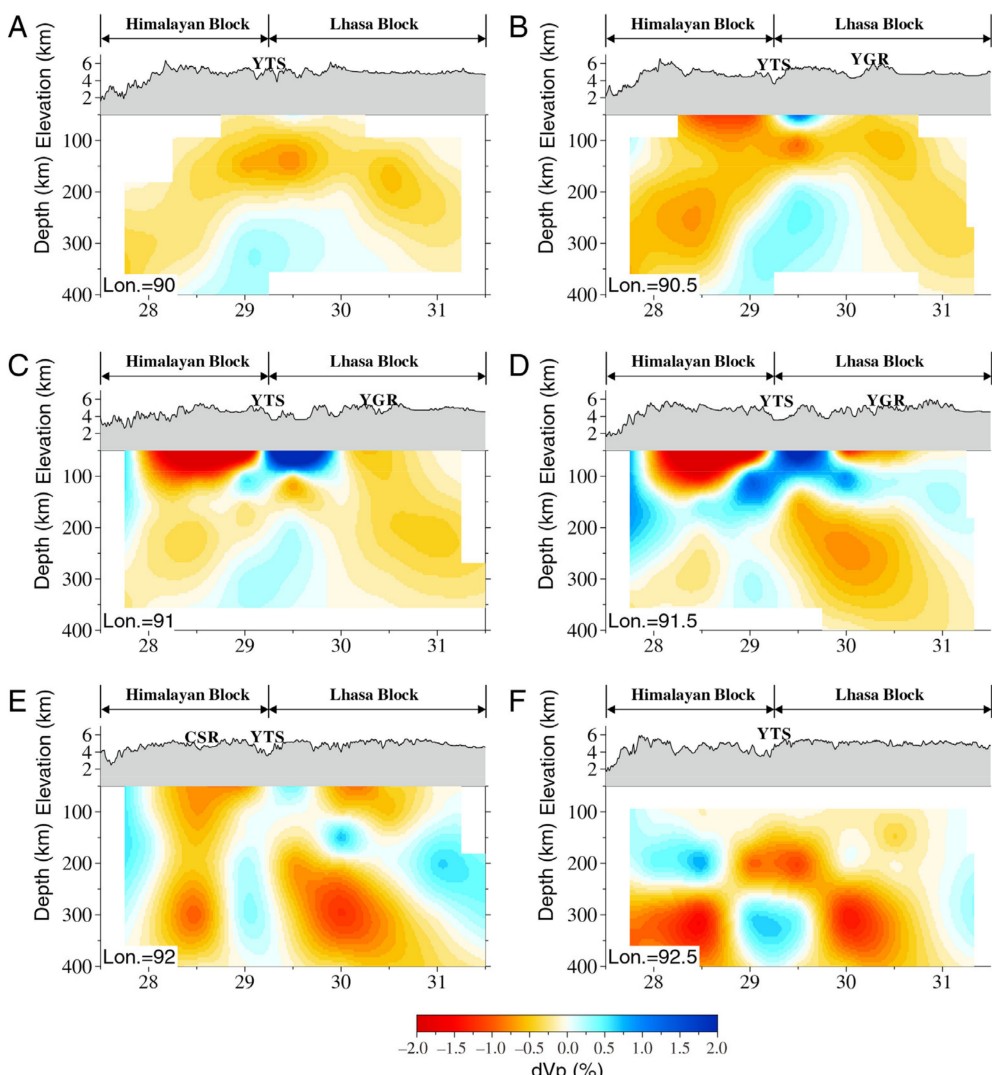

**Figure 9.** (**A**–**F**) North–south vertical cross-sections of the 3D P-wave velocity (Vp) model along six profiles whose longitude is shown at the lower-left corner of each panel. The red and blue colors represent low and high Vp perturbations, respectively, whose scale is shown at the bottom. The major tectonic blocks are displayed above each panel.

### 3.4. Restoring Resolution Tests

We conducted the restoring resolution test (RRT) to further assess the major characteristics of our tomographic result and the obtained Vp model was constructed as the synthetic input model. Random noise with a standard deviation of 0.1 s was added to the synthetic arrival-time data. Then, we conducted the tomographic inversion with the same ray paths and methods as the observed data. In general, the RRT results showed that the input model can be well recovered beneath most parts of the study region at depths above ~400 km, albeit that the amplitude of some input Vp anomalies in the western part of the study region reduced slightly due to the sparse ray coverage (Supplementary Figure S4B–D). In general, the restoring resolution tests showed that the synthetic Vp anomalies beneath the study region were well resolved and our Vp models were quite robust.

### 3.5. Tectonic Implications

Alternating low- and high-Vp anomalies were observed beneath the Himalayan and Lhasa blocks across the YTS, indicating that strong lateral heterogeneities exist in the upper mantle beneath the study region. A prominent LVZ extending to ~150 km depth

was observed beneath the Himalayan block to the south of the YTS (Figure 8A,B and Figure 9C,D), which might be related to the fragmentation of the ICL and induce localized upwelling of the asthenospheric materials from the upper mantle. There exists a significant HVZ beneath the Lhasa block beside the LVZ (Figure 8A,B and Figure 9C,D), which might indicate the rigid and remnant Tibetan lithosphere basement. In addition, relatively prominent high-Vp anomalies at depths of 300–400 km were visible beneath the YTS (Figure 9A–C), which might be associated with the delamination of the underthrusting ICL due to the resistance of the rigid Lhasa block. There exist relatively significant high-Vp anomalies beneath the Himalayan block dipping northward with a relatively flat angle at depths of 100–200 km (Figure 9D,F), which might indicate the subducted ICL. In contrast, a relatively prominent high-Vp anomaly with high dipping angle was observed at 200–300 km depths beneath the northern edge of the Lhasa block near the BNS (Figure 9E), which was generally consistent with the previous body-wave tomographic result [18] and can be interpreted as the subducting ICL. Significant low-Vp anomalies were observed in the upper mantle extending to ~300 km depth beneath the YGR and the CSR (Figure 8B–E and Figure 9B–E), which were in good agreement with previous tomographic results [17,18], indicating that the north–south trending rifts might cut through the entire lithosphere and serve as channels for the asthenospheric materials migrating upward to the surface, which is also supported by the widespread distributions of ultra-potassic volcanic rocks along the rifting zones [59]. The low-Vp anomaly beneath the YGR is connected to the LVZ close to the YTS (Figure 9B,C), and the previous study suggests that there exists a seismicity gap beneath the YGR, suggesting that the northward subducted Indian lithosphere has been torn there and is featured as a weak zone [60]. The tearing of the ICL is also supported by previous SKS-wave splitting and receiver-function studies [61,62]. In addition, the previous study shows that intermediate-depth earthquakes exhibit concentrated distributions in the lower crust and upper mantle beneath several areas of southern Tibet, indicating that the ICL has been torn into multiple pieces with different angles [60]. Combining our 3D Vp model with many previous results, we present a cartoon to describe the lithospheric structure beneath the YTS and the formation mechanism of major rifting systems in the study region (Figure 10). We deem that the complex deep structures of southern Tibet might be associated with various geodynamic mechanisms rather than any single tectonic event.

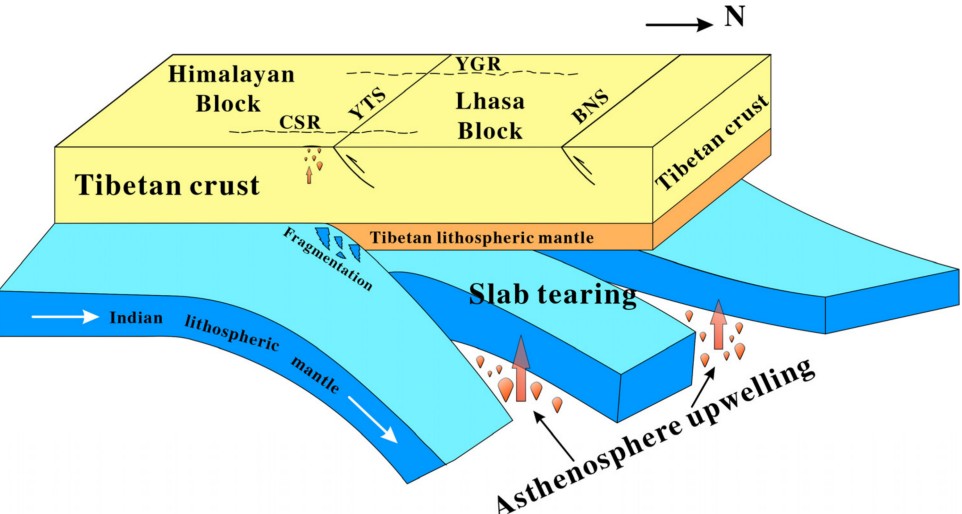

**Figure 10.** A cartoon showing the lithospheric structure beneath the southern Tibet influenced by the subduction of the Indian continental lithosphere. YTS: Yarlung–Tsangpo suture; BNS: Bangong–Nujiang suture; YGR: Yadong–Gulu rift; CSR: Cona–Sangri rift.

## 4. Conclusions

In this study, we determined detailed 3D Vp structures of the upper mantle extending to ~400 km depth beneath the YTS by using 495 high-quality teleseismic events recorded at 20 three-component broadband seismographs deployed for ~20 months. Our 3D Vp models exhibited higher lateral resolution in comparison to previous tomographic results and revealed more small-scale lateral variations of the upper-mantle velocity structures beneath the Himalayan and Lhasa blocks across the YTS. The main results of this work are summarized as follows:

(1) There exists a prominent HVZ beneath the southern edge of the Lhasa block above ~100 km depth, indicating that a rigid lithosphere basement may still remain.

(2) A significant LVZ is visible beneath the Himalayan block, which could be related to the fragmentation of the underthrusting ICL due to the resistance of the rigid Lhasa block, resulting in small-scale asthenospheric material upwelling.

(3) The lateral variations in the subduction angles of the Indian lithosphere could contribute to the slab tearing along active rifting systems.

(4) The YGR and CSR might serve as channels for the upwelling of asthenospheric materials, which could play an important role in the tectonic evolution of the Himalayan–Tibetan orogen.

**Supplementary Materials:** The following supporting information can be downloaded at: https://www.mdpi.com/article/10.3390/rs15112724/s1. Figure S1: The 1D reference velocity model adopted in the tomographic inversion. Figure S2: Map views showing results of a checkerboard resolution test (CRT) with a lateral grid interval of 0.35°. The layer depth is shown at the upper-left corner of each map. The open and solid circles denote positive and negative Vp perturbations (%), respectively, whose scale is shown at the bottom. Other labels are the same as those in Figure 4. Figure S3: The same as Figure S2 but with a lateral grid interval of 0.75°. Figure S4: North–south vertical cross-sections showing results of the restoring resolution test (RRT) along six profiles whose longitude is shown at the lower-left corner of each panel. The red and blue colors represent low and high Vp perturbations, respectively, whose scale is shown at the bottom.

**Author Contributions:** Conceptualization, Y.T.; methodology, D.Y. and Y.T.; formal analysis, D.Y., Y.T. and Z.L.; investigation, Z.L. and H.L.; data curation, D.Y. and Y.T.; writing—original draft preparation, D.Y.; writing—review and editing, Y.T. and H.L.; funding acquisition, Y.T. All authors have read and agreed to the published version of the manuscript.

**Funding:** This study is supported by the Second Tibetan Plateau Scientific Expedition and Research Program (STEP), (Grant No. 2019QZKK0701), the National Natural Science Foundation of China (Grant No. 42274065), the Program for JLU Science and Technology Innovative Research Team (No. 2021TD-05), and Fundamental Research Funds for the Central Universities in China.

**Data Availability Statement:** The arrival-time data used for this study and the 3D Vp model are archived at the website: https://doi.org/10.6084/m9.figshare.22742993.

**Acknowledgments:** We truly appreciate the editor and three anonymous reviewers for their constructive review comments and suggestions, which have improved this paper. We thank Zhao at Tohoku University for providing the seismic tomography code (TOMO3D) used in this study. The Generic Mapping Tools (GMT) software package was used to plot most of the figures [63].

**Conflicts of Interest:** The authors declare no conflict of interest.

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
