# Peer review of "Upper Mantle Velocity Structure Beneath the Yarlung–Tsangpo Suture Revealed by Teleseismic P-Wave Tomography"

_remotesensing, doi:10.3390/rs15112724_

Round 1
Reviewer 1 Report
This is an interesting study on the upper mantle P-wave velocity structure beneath southern Tibet. However, I have a few comments that should be considered prior to publication.
• The font of the coordinate axis in Figure 3B and 3C should be consistent.
• In the section “3.3. Tomographic Imaging”, please emphasize some new features obtained by this study which is different from or much clearer than previous tomographic results.
• It seems that the figures in this manuscript are plotted by the GMT (Generic Mapping Tools) software. Please cite any reference in the acknowledgements section.
Lines 17-18: Please modify to “which significantly improves the efficiency and precision of…”
Line 25: underthrusing→underthrusting
Line 26 and line 275: astheospheric→asthenospheric
Lines 29-30: Please modify to “which contributes to the widespread distribution of…”
Lines 50-51: Please modify to “result in widespread post-collisional potassic volcanism…”
Line 54: Indian shield→Indian Shield
Line 82 and line 86: underthrusing→underthrusting
Line 86: remains→remain
Line 112: teleseimic→teleseismic
Line 144: indicate→indicates
Line 201: horizonal→horizontal
Line 212: part→parts
Line 213: interval→intervals
Line 276: besides→beside
Author Response
Comments and Suggestions for Authors
This is an interesting study on the upper mantle P-wave velocity structure beneath southern Tibet. However, I have a few comments that should be considered prior to publication.
Thank you for your comments on our manuscript. These comments are precious and helpful for great improvements to the manuscript. In this revised version, each of the comments has been carefully addressed, and changes made for corrections are marked up using the “Track Changes” function in the manuscript.
- The font of the coordinate axis in Figure 3B and 3C should be consistent.
Yes, we have revised it. Please see Figure 3 of the revised manuscript.
- In the section “3.3. Tomographic Imaging”, please emphasize some new features obtained by this study which is different from or much clearer than previous tomographic results.
Nice point! Please see lines 234-243 of the revised manuscript.
- It seems that the figures in this manuscript are plotted by the GMT (Generic Mapping Tools) software. Please cite any reference in the Acknowledgements section.
Done. Please see lines 343-344 of the revised manuscript.
Comments on the Quality of English Language
Lines 17-18: Please modify to “which significantly improves the efficiency and precision of…”
Yes, we have revised it.
Line 25: underthrusing→underthrusting
Done.
Line 26 and line 275: astheospheric→asthenospheric
Done.
Lines 29-30: Please modify to “which contributes to the widespread distribution of…”
Yes, we have revised it.
Lines 50-51: Please modify to “result in widespread post-collisional potassic volcanism…”
Yes, we have revised it.
Line 54: Indian shield→Indian Shield
Done.
Line 82 and line 86: underthrusing→underthrusting
Done.
Line 86: remains→remain
Done.
Line 112: teleseimic→teleseismic
Done.
Line 144: indicate→indicates
Done.
Line 201: horizonal→horizontal
Done.
Line 212: part→parts
Done.
Line 213: interval→intervals
Done.
Line 276: besides→beside
Done.
We have corrected all the above-mentioned spelling mistakes in the revised manuscript. Thank you very much.

Reviewer 2 Report
Overall a good case study to build high-resolution velocity model with teleseismic data.
The authors use one 2-D array for teleseismic imaging. However the area do exist more array data and may provide better resolution coverage if more data can be used. Explain the reason of using only one array in this study.
English writing can be improved over all, especially in the introduction and conclusion. Spelling mistakes should be checked throughout whole text, for example 'underthrusing' in line 84.
Author Response
Comments and Suggestions for Authors
Overall a good case study to build high-resolution velocity model with teleseismic data.
Thank you very much.
The authors use one 2-D array for teleseismic imaging. However, the area does exist more array data and may provide better resolution coverage if more data can be used. Explain the reason of using only one array in this study.
Nice point! The permanent seismic stations in the present study region are scarce and thus we did not use them in case resulting in a lower resolution if we expanded the study area. We will collect more data from portable seismic stations within the study region in follow-up studies to further improve the resolution of our P-wave velocity model.
Comments on the Quality of English Language
English writing can be improved over all, especially in the introduction and conclusion. Spelling mistakes should be checked throughout whole text, for example 'underthrusing' in line 84.
Yes, we have improved the English writing and checked the spelling mistakes throughout the whole text of the revised manuscript.

Reviewer 3 Report
The paper is of good scientific level and practical interest. The structure of the article is well organized, the text is clear, experimental studies are used correctly.
Technical remarks include: 1) the presence of blue figures in Fig.1; 2) there is no explanation of what the numbers mean, for example (0-90), in the lower left corner of each segment in Fig. 4; 3) In Fig. 10, for the Indian continental lithosphere, it is necessary to show the mantle lithosphere, as was done for the Tibet.
The article can be published in its current form after the removal of minor technical errors.

Author Response
The paper discusses the results of teleseismic tomography to investigate the 3-D P-wave velocity (Vp) structure of the crust and upper mantle at depths of 50-400 km beneath the Yarlung-Tsangpo Suture (YTS). The results obtained show that alternating low- and high-Vp anomalies are visible beneath the Himalayan and Lhasa blocks across the YTS, indicating that strong lateral heterogeneities exist beneath the study region. A significant high-Vp zone is visible beneath the southern edge of the Lhasa block at 50-100 km depths close to the YTS, which might indicate the rigid Tibetan lithosphere basement. There exists a prominent low-Vp zone beneath the Himalayan block to the south of the YTS extending to ~150 km depth, which might be associated with the fragmentation of the underthrusting Indian continental lithosphere (ICL) and induce localized upwelling of asthenospheric materials from the upper mantle. In addition, significant low-Vp anomalies are observed beneath the Yadong-Gulu Rift and the Cona-Sangri Rift extending to ~300 km depth, indicating the tearing of the subducted ICL might provide pathways for the localized asthenospheric materials upwelling, which contribute to the widespread distributions of north-south trending rifts and geothermal activities in the southern Tibet.
Thank you for your comments on our manuscript. These comments are precious and helpful for great improvements to the manuscript. In this revised version, each of the comments has been carefully addressed, and changes made for corrections are marked up using the “Track Changes” function in the manuscript.
The paper is of good scientific level and practical interest. The structure of the article is well organized, the text is clear, experimental studies are used correctly. Technical remarks include:
1) the presence of blue figures in Fig.1;
The blue marks in Figure 1 denote the main lakes in the study region. We have removed them in Figure 1 of the revised manuscript.
2) there is no explanation of what the numbers mean, for example (0-90), in the lower left corner of each segment in Fig. 4;
Yes, we have explained the meaning of numbers in the lower-right corner of each map. Please see lines 157-158 of the revised manuscript.
3) In Fig. 10, for the Indian continental lithosphere, it is necessary to show the mantle lithosphere, as was done for the Tibet.
Yes, we have revised it. Please see Figure 10 of the revised manuscript.
The article can be published in its current form after the removal of minor technical errors.
Thank you very much.
